# A Novel Approach for Identifying Nanoplastics by Assessing Deformation Behavior with Scanning Electron Microscopy

**DOI:** 10.3390/mi14101903

**Published:** 2023-10-05

**Authors:** Jared S. Stine, Nicolas Aziere, Bryan J. Harper, Stacey L. Harper

**Affiliations:** 1School of Chemical, Biological and Environmental Engineering, Oregon State University, Corvallis, OR 97331, USA; stinej@oregonstate.edu; 2Department of Environmental and Molecular Toxicology, Oregon State University, Corvallis, OR 97331, USA; bryan.harper@oregonstate.edu; 3School of Electrical Engineering and Computer Science, Oregon State University, Corvallis, OR 97331, USA; azieren@oregonstate.edu

**Keywords:** SEM, detection, microplastics, machine learning, polymers

## Abstract

As plastic production continues to increase globally, plastic waste accumulates and degrades into smaller plastic particles. Through chemical and biological processes, nanoscale plastic particles (nanoplastics) are formed and are expected to exist in quantities of several orders of magnitude greater than those found for microplastics. Due to their small size and low mass, nanoplastics remain challenging to detect in the environment using most standard analytical methods. The goal of this research is to adapt existing tools to address the analytical challenges posed by the identification of nanoplastics. Given the unique and well-documented properties of anthropogenic plastics, we hypothesized that nanoplastics could be differentiated by polymer type using spatiotemporal deformation data collected through irradiation with scanning electron microscopy (SEM). We selected polyvinyl chloride (PVC), polyethylene terephthalate (PET), and high-density polyethylene (HDPE) to capture a range of thermodynamic properties and molecular structures encompassed by commercially available plastics. Pristine samples of each polymer type were chosen and individually milled to generate micro and nanoscale particles for SEM analysis. To test the hypothesis that polymers could be differentiated from other constituents in complex samples, the polymers were compared against proxy materials common in environmental media, i.e., algae, kaolinite clay, and nanocellulose. Samples for SEM analysis were prepared uncoated to enable observation of polymer deformation under set electron beam parameters. For each sample type, particles approximately 1 µm in diameter were chosen, and videos of particle deformation were recorded and studied. Blinded samples were also prepared with mixtures of the aforementioned materials to test the viability of this method for identifying near-nanoscale plastic particles in environmental media. Based on the evidence collected, deformation patterns between plastic particles and particles present in common environmental media show significant differences. A computer vision algorithm was also developed and tested against manual measurements to improve the usefulness and efficiency of this method further.

## 1. Background

In recent years, studies focusing on the extent of global plastic pollution, specifically micro and nanoplastic (MNP) pollution, have increased exponentially [1]. With this increase in research, mounting evidence of the ubiquity of MNPs in the environment has raised concerns over their potential implications for the health of terrestrial and aquatic ecosystems [2,3,4,5]. Microplastics are commonly described as plastic particles less than 5 mm in size, while nanoplastics have been described as having at least one dimension smaller than 1000 nanometers (nm) [6]. The smallest nanoplastics are of particular concern due to their increased capacity for biological interactions [5,7,8]. Given the difficulty of measuring or identifying nanoplastics from environmental samples due to their small size and low mass, there remains a methodological gap in characterizing environmental nanoplastics [6,9]. New, accessible approaches for detecting nanoplastics would greatly improve our understanding of their presence in the environment and provide additional data for regulatory decision-makers.

Following concentration and recovery from environmental samples, analysis using tools like pyrolysis gas chromatography–mass spectrometry (Py-GC-MS) may support the identification of nanoplastics by polymer type, but quickly detecting environmentally relevant concentrations of nanoplastics may prove a challenge as limits of detection are on the order of micrograms per liter range [6,10,11]. In addition to the significant preconcentration needed to enable the effective use of Py-GC-MS [12], it also requires the destruction of a sample, inhibiting the collection of nanoplastic morphology or particle count data once analyzed. The typical non-destructive methods used for collecting chemical fingerprints to identify microplastics by polymer type, including Fourier-transform infrared and Raman spectroscopy, have been reported to have limitations due to small particle size and background interference when used to characterize nanoplastics [9]. A recent study focused on applying scanning transmission X-ray spectromicroscopy (STXM) and near-edge X-ray absorption fine-structure spectroscopy (NEXAFS) to image and characterize spiked nanoplastics recovered from different environmental matrices [13]. While this method seems viable for its intended purpose, it still requires considerable time to image particles and perform spectral analysis. For environmental nanoplastics research to advance more rapidly, simpler and more accessible methods are needed.

Methods targeting unique molecular structures of different polymers may provide an analytical fingerprint associated with individual polymer types, allowing researchers to better characterize nanoplastics. In electron energy loss spectroscopy (EELS), materials are exposed to an electron beam with a known energy input, while energy losses resulting from inelastic scattering of electrons are measured to create spectra unique to a given material [9,14]. Similar techniques commonly coupled with SEM, including X-ray photoelectron spectroscopy (XPS) and energy-dispersive X-ray spectroscopy (EDS), have also been applied while attempting to identify MNPs within a sample. One study investigating Indian beach sediment utilized both SEM and EDS to verify the presence of PVC, PE, and PET microplastics [15]. However, the microplastics identified using this methodology were between 36 μm and 5 mm in size and were identified primarily based on strong carbon signatures in their elemental spectra. This methodology alone would likely prove more challenging for characterizing nanoplastics in samples of mixed environmental media with other high-carbon signature materials. Researchers using these techniques to identify MNPs typically rely on heavier elemental signatures as markers that may not always be present or known to be unique to plastics [15,16,17]. 

Due to the intensity of the SEM electron beam, many organic and sensitive samples may degrade during imaging if left untreated [18,19]. Often, it is desirable to coat sensitive samples with thin carbon or gold–palladium coatings to help protect the sample from radiation damage incurred by the electron beam [20]. If left uncoated, organic materials such as polymers with relatively low thermal conductivity values are susceptible to deformation with only moderately elevated temperatures [21]. By maintaining constant electron beam parameters during SEM imaging, it may be possible to identify unique deformation profiles for anthropogenic plastics at the nanoscale that are distinct from other environmental media. Furthermore, utilizing SEM enables the observation of materials down to the nanoscale, which is outside the detection limits for many other analytical methods.

Electron-beam irradiation is commonly used for sterilizing ultra-high molecular weight polyethylene (UHMWPE) materials used in biomedical applications [22,23]. Irradiation of linear hydrocarbon polymers can result in C-C and C-H bond cleavage, radical and hydrogen removal, chain scission, cross-linking, and oxidation (in the presence of oxygen) [22]. Polymer research has shown that chemical cross-linking is an irreversible process that is commonly used to improve the strength, stiffness, and rigidity of polymeric materials [24]. These chemical alterations, occurring concurrently with high enough irradiation doses, can lead to spatial rearrangement and complex physical changes in polymer characteristics [23]. During SEM, this spatial rearrangement of the polymer matrix can result in changes to polymer surface characteristics that are observable in real time. By controlling the conditions of irradiation during SEM, we hypothesized that physical alterations of MNPs of similar sizes from the most common commercial plastics would occur predictably by polymer type and be unique from the deformation patterns seen in non-anthropogenic polymers. This study sought to test the capacity for irradiation of individual particles during SEM to provide a spatiotemporal fingerprint to identify environmental MNPs.

A supporting computer vision (CV) algorithm has been written in Python using the popular PyTorch application processing interface (API) to aid the processing power of the developed methodology [25]. PyTorch contains tools allowing software development to design and train deep neural networks. The dataset of videos collected with SEM, along with their associated masks representing particles to segment, were used as training data. Using a specifically designed objective function, the network was trained to accurately predict pixels belonging to either the foreground or background. The architecture, the objective function, and the training process were all programmed with PyTorch. Once the deep neural network was trained, it was stored in memory and used to infer a segmentation mask for new input data. The prediction quality was quantitatively evaluated using a standard metric for the segmentation algorithm, namely intersection over union (IoU). IoU quantifies the amount of overlap between the predicted and manually annotated masks. A 100% IoU corresponds to a predicted mask perfectly aligned with the manually annotated one. As artificial intelligence (AI) systems continue to develop and increase in sophistication, the accuracy of automated analysis of data collected using the methodology described in this study is expected to improve.

## 2. Materials and Methods

### 2.1. Materials and Characterization

Environmentally relevant polymer types were investigated by identifying the most prominent commercial plastics found in environmental waste [26]. After reviewing the fundamental properties of engineered polymers, the degree of crystallinity was noted to be affected during SEM irradiation [23,27] and hypothesized to be the most predictive of polymer deformation during irradiation. Considering the predicted deformation profiles of different polymer types during SEM imaging, the degree of crystallinity values and overall polymeric structure were used to identify polymer types anticipated to capture a wide spectrum of particle deformation behavior. The three plastic materials chosen for this study were polyvinyl chloride (PVC), polyethylene terephthalate (PET), and high-density polyethylene (HDPE). Pristine samples of each chosen polymer type were selected from the Hawaii Pacific University Center for Marine Debris Research Polymer Identification Kit and individually fragmented using a Retsch Cryomill to generate environmentally relevant MNPs for SEM analysis.

Non-plastic materials were also studied for comparison against MNPs to determine if plastics deform differently under SEM irradiation analysis. Non-plastic materials were selected to capture a range of media commonly found in environmental samples. Algae were selected as a proxy for common biological material. Samples of algae (*Raphidocelis subcapitata*) were prepared using specimens cultured within the laboratory. For a non-polymeric material, aluminum silicate (kaolinite) was selected as a proxy for soft silt and sedimentary particles commonly found in environmental samples. Kaolinite materials were obtained through Sigma–Aldrich. Environments are also rich in natural polymeric materials, and a naturally derived polymer would also be needed to compare against the anthropogenic polymers in this study. Cellulose was selected as a proxy for naturally occurring polymer materials common in environmental media and could be mistaken as an MNP. Cellulose used in this study was obtained through Sigma–Aldrich.

### 2.2. SEM Sample Preparation

All SEM sample preparation occurred within a laminar flow hood with materials obtained through Ted Pella, Inc. (Redding, CA, USA). Aluminum SEM specimen mounts were first prepared by placing a piece of double-stick carbon tape and affixing a 5 × 5 mm silicon wafer to the top of the tape. Compressed air was then used to remove any potential dust or particulate debris that may have been present on the sample. Plastic samples were prepared by suspending milled plastics in ultrapure water and subsequently, using glass pipette tips, drop-casting each sample onto individual specimen mounts. Both kaolinite and nanocrystalline cellulose samples were prepared similarly by suspending the dry powder materials in ultrapure water and subsequently drop casting onto individual SEM specimen mounts. Since algae samples were already suspended in aqueous media, they were diluted 50:50 with ultrapure water to reduce the concentration of algal cells prior to drop casting. Samples were left uncoated to enable observation of deformation during SEM irradiation. All validation samples were prepared similarly using mixtures of plastic materials and environmentally relevant media.

### 2.3. Experimental Design

While field emission SEM (FE-SEM) may also be suitable for obtaining high-resolution images of MNPs at low voltages, the higher voltages desired and the general ease of accessibility from a typical SEM were preferable. To ensure consistent energy input across multiple particle deformation observations, electron beam voltage and current were maintained at 5 kilovolts (kV) and 33 nanoamperes (nA), respectively, with the beam aperture set to 50 µm. The beam scan rate was held constant at 500 nanoseconds with a horizontal field width (HFW) of 9.95 µm and a working distance of 10.6 mm for each sample. For the purposes of method development, groups of five of the smallest particles of each material type (between 1 and 10 µm in length) were selected, and videos of particle deformation were recorded for at least 40 s for analysis using a Quanta 3D dual beam SEM (FEI Company, Hillsboro, OR, USA). Images were automatically collected every second during recording for particle deformation analysis.

Electron beam settings were held constant throughout all plastic sample observations for PVC, PET, and HDPE. After the initial observations of the selected plastic materials, it was decided that a higher beam current may help further interrogate differences between the deformation behavior of particle materials. All subsequent algae, kaolinite, and cellulose observations occurred with a beam voltage of 5 kV at a current of 37.9 nA, with all other experimental parameters remaining the same. Additional observations of different particles from the PVC, PET, and HDPE samples were also recorded at the higher beam current to determine potential changes in observed deformation patterns resulting from the increased SEM irradiation.

Following the deformation observations of all the materials used in this study, three blinded validation samples were prepared using mixtures of the same materials with the addition of environmentally relevant media. These samples were then observed under the same electron beam parameters with the increased 37.9 nA beam current. Particles present in the validation samples were identified systematically prior to observation. Validation samples were divided into quadrants during SEM analysis, with 10 particles of similar size from each quadrant being selected and recorded under SEM irradiation. In total, 40 randomly selected particles from each of the three blinded samples were recorded for deformation analysis. Deformation profiles from these blinded particles were analyzed and compared against the deformation profiles collected for the six known materials used in this study. The intent of the blinded validation study was to evaluate the utility of this method for detecting MNPs from complex environmental samples.

### 2.4. Manual Data Evaluation

Analysis of the change in cross-sectional areas of individual particles was the primary focus of this study. Once the SEM irradiation observations were recorded for particle groups for each sample material, particle cross-sections were measured using images taken at specific time points. Particle cross-sectional area measurements were collected using ImageJ Version 1.53i image processing software (NIH, Madison, WI, USA). As the plastic particles were of irregular size and shape due to milling, variability among particle deformation within individual polymer samples was expected. Following the characterization of deformation profiles for known materials, particle cross-section measurements from blinded samples were then collected. After the characterization of deformation profiles for particles of unknown origin, these data were compared against those of the known materials. If the unknown particles exhibited similar deformation profiles to those of known plastics, this would indicate potential MNPs present in the blinded samples. 

Once the analysis of particles from blinded samples was complete and deformation profiles were analyzed, the presence of MNPs in blinded samples was proposed and validated by the person who prepared the blinded samples. Successful identification of MNPs present in mixed samples would further validate this method and indicate its potential to help close the methodological gap for identifying environmental nanoplastics. However, the method described herein for manual cross-sectional particle area measurements does not lend itself to the practical and rapid collection of environmental data on MNPs. More automated measurement techniques would be needed to develop this method further.

### 2.5. Automated Data Evaluation with Computer Vision Analysis

Automated data evaluation was developed to expand the usefulness of the developed methodology. By developing a computational tool that can observe the SEM irradiation of particles and calculate the changes in particle size, particle deformation profiles could be produced in a fraction of the time.

Computer vision (CV) systems are proper candidates to tackle the problem of measuring changing particle sizes within the observational data collected in this study. Given an image representing a particle, the task consists of recognizing which pixels represent it. This is known as the segmentation task, a popular problem that has been widely studied over the years by the CV community. Traditional segmentation algorithms may be used, including watershed [28], grab-cut [29], or image preprocessing followed by thresholding. However, these methods suffer from poor generalization power, are sensitive to noise, and require tedious manual tuning of parameters. They become poor candidates for SEM image processing, which can be highly noisy with varying contrasting occurring. With the recent advance of AI systems, particularly deep learning, a new set of algorithms was developed, leveraging deep neural networks’ discrimination power. This family of techniques achieves state-of-the-art performance in various CV tasks, like segmentation.

Deep learning algorithms must be trained on a large set of annotated data to perform well on the desired downstream task. Since the particle data collected were not annotated with the corresponding image masks representing the particle, selecting a database for training the deep network with publicly available annotated data was necessary. The database selected is called PhC-C2DH-U373 [30] for cell segmentation. This dataset is appropriate for training the deep network because the images are annotated with corresponding expert-made segmentation masks, and the cells represented are visually similar to the particles observed in this study. The visual domain is also similar since both databases contain images taken using SEM. The deep network model comprises the popular UNet architecture [31]. UNet is a popular choice for the segmentation algorithm because it was designed to consider the image at multiple scales and is robust to noise perturbations. It achieves state-of-the-art performance on multiple benchmark datasets on the segmentation task.

The first and last frames of all particle deformation videos were manually annotated to fill the gap between the cell and particle deformation datasets. UNet architecture was then trained on a joint set of images with cell and plastic deformation masks, increasing its ability to generalize to unseen images containing plastic particles.

### 2.6. Statistical Analysis

Statistical analyses were performed using SigmaPlot version 15.0 (Systat Software, San Jose, CA, USA). Differences between sample groups were considered significant when *p* ≤ 0.05. Significant differences in deformation behavior between material types were based on cross-sectional measurement data collected over time and analyzed using a two-way repeated-measures analysis of variance (RM-ANOVA) and Bonferroni post hoc analysis. Significant differences in initial deformation rate data across material types were determined using one-way ANOVA and Tukey’s post hoc analysis. Normality and equal variance of data were determined using Shapiro–Wilk and Brown–Forsythe tests, respectively. Correlations between variables within the study were also performed using linear regression analysis.

## 3. Results

### 3.1. Plastic Particle Deformation

Data collected for PVC, PET, and HDPE particles at a 33 nA beam current showed a variation in deformation patterns between the different polymer types. Figure 1 shows particle deformation over time as a percentage of the initially measured cross-sectional area. The data shown are the mean of five different particle measurements for each polymer type (*n* = 5), with standard error bars showing the deformation variability between measured particles. Trends in the data show that the measured particle cross-sectional area is generally reduced for the lower crystallinity polymer types during SEM irradiation. Although distinct differences between particle deformation seem apparent, the higher variability in PVC particle deformation adds uncertainty to the dataset. Statistical comparisons of the three deformation profiles show that PVC and PET deformation profiles significantly differed from HDPE (*p* < 0.001) but not from each other.

Following initial plastic particle deformation observations, additional data on plastic materials were collected at a higher beam current of 37.9 nA to observe changes resulting from increased irradiation. PVC, PET, and HDPE were re-evaluated by selecting five additional particles within each sample for analysis (*n* = 5). Data collected for all plastic materials under both 33 nA and 37.9 nA beam currents are shown in Figure 1. Statistical comparisons of the deformation profiles between the plastic materials at different beam currents indicated significantly increased deformation behavior for PVC at a higher beam current (*p* = 0.042) but no significant change for either PET or HDPE. The apparent reduced average deformation of PET particles observed at the 37.9 nA beam current is likely an artifact of variations in the degree of crystallinity across the particles selected in the different study groups. Comparisons across the three materials at the higher beam current also indicate that differences in deformation behavior between all of the plastic materials were statistically significant from each other. Based on these observations, deformation profiles of lower crystallinity plastics under higher currents appear more likely to display accelerated deformation rates and greater changes in measured cross-sectional area than higher crystallinity plastics. Linear regression analysis also showed that the final deformation measurements for each material (*t* = 39 s) correlated to the reported degree of crystallinity values with an R^2^ value of 0.87.

### 3.2. Plastic vs. Non-Plastic Media Particle Deformation

Generally, data collected for algae, kaolinite, and cellulose particles indicated less deformation than plastic particles. Notable differences in particle morphology were also present during deformation observations. Interestingly, blebbing was observed during algal cell irradiation, which enabled additional qualitative distinction of algal media from other media in this study. Kaolinite was characterized by a markedly different contrasting quality over the other materials and appeared to have more jagged features when compared to the other materials. While cellulose appeared to have a similar morphology to the plastic materials observed in this study, it did not appear to degrade as readily as the lower crystallinity plastics in this study. Even at the higher current of 37.9 nA, all of the non-plastic media tested appeared to display less particle shrinkage when compared to both PVC and PET samples. 

Figure 2 shows particle deformation profiles for algae, kaolinite, and cellulose, as well as the plastic materials tested at 37.9 nA beam current. Data plotted in Figure 2 are the mean of five individual particle measurements for each material type (*n* = 5) collected during SEM irradiation with calculated standard error bars to indicate variability within material types. Measurements were then normalized to a percentage of the initially measured cross-sectional area over time. Statistical comparisons across the material types shown indicate that both PVC and PET display deformation behavior significantly different from the rest of the materials tested and from each other (*p* < 0.001). Additional comparisons yielded no significant differences in particle deformation behavior between the non-plastic media or HDPE. These observations further indicate that using this methodology, the particle deformation behavior of low crystallinity plastics is significantly different from the behavior of common non-plastic environmental media and high crystallinity plastics. SEM images showing typical particle deformation for each material type are shown in Figure 3.

### 3.3. Blinded Validation Sample Analysis

Measurements were collected at five time points for 0, 2, 4, 10, 20, and 40 s of irradiation to more rapidly collect the deformation profiles for particles from blinded samples. Deformation profiles were then plotted and visually assessed to determine if any particles of unknown origin exhibited deformation behavior similar to the lower crystallinity plastics in this study (PVC and PET). Similar particle shrinkage behavior to these known plastics was considered indicative of anthropogenic polymers, and those particles were further studied. Unknown particles were deemed suspect when their overall particle deformation at the end of observation (*t* = 39 s) showed deformation greater than the least deforming known PET particles assessed in this study (8.1% at *t* = 39 s). 

Based on analyses of the 120 unknown particles studied across the three blinded validation samples, 35 unknown particles exhibited suspect deformation behavior. Of the total suspect particles, 19, 1, and 15 were present in blinded validation samples #1, #2, and #3, respectively. Suspect particles in each validation sample were then grouped and compared for statistical similarity to known deforming plastics PVC and PET. Upon comparison, it was found that PVC was significantly different from the grouped suspect particles in all three validation samples (*p* < 0.002), whereas no significant differences were apparent between PET and any of the three validation samples. These data indicated the possible presence of PET in every blinded validation sample. 

To further interrogate the presence of PVC within the validation samples, a new set of suspect unknown particles were grouped based on overall particle deformation at the end of observation (*t* = 39 s) and showed deformation greater than the least deforming known PVC particle assessed in this study (20.0% at *t* = 39 s). Of the total suspected PVC particles, 3, 0, and 2 were present in blinded validation samples #1, #2, and #3, respectively. After a statistical comparison, PVC was no longer found to be statistically different from validation sample #3. This analysis indicated that the suspected particles in validation sample #1 exhibited deformation profiles that were different from those of the suspected particles in validation sample #3. Additionally, this analysis showed no statistical difference between PVC and validation sample #3, indicating the potential presence of PVC. Following these analyses predicting the presence of PET in all validation samples and PVC in validation sample #3, the data were verified with the preparer of the blinded samples for comparison. The results of those predictions are highlighted in Table 1.

### 3.4. Analysis of Materials with AI-Assisted Data Processing

The same particle observation data analyzed with manual measurements was also analyzed using the developed machine learning algorithm for comparison to provide enhanced data-generating power. The benefits of machine learning analysis are exceptionally enhanced speed of data collection while also collecting data at additional time points. Figure 4 depicts the computationally generated particle measurements for the plastic materials at the lower 33 nA beam current. Statistical analysis of the three plastic materials at a 33 nA beam current showed PVC to be significantly different from HDPE (*p* = 0.005), but PET was found not to be significantly different from HDPE (*p* = 0.088). PVC and PET were also not found to be significantly different from each other. These results deviate slightly from those derived from the manual measurements, possibly suggesting the need for further training of the machine learning algorithm to improve the accuracy of measurements and reduce the variability of the data collected.

Figure 5 depicts computationally generated particle measurements for all materials tested at the 37.9 nA beam current. Upon visual inspection, the same trends in the data are apparent, albeit with increased variability. Statistical comparisons across the material types indicated that PVC displays deformation behavior significantly different from all the other materials except for PET (*p* = 0.07). Given the high variability of the PET data generated, PET was no longer shown to be significantly different from the other materials, including HDPE (*p* = 0.064). These results also deviate slightly from those derived from the manual measurements. A comparison of manual and computationally generated particle measurements is detailed below in Table 2. Each material type was assessed to determine significant differences between manual and computationally generated data. 

The trends between manual and computationally generated measurements are similar, such as no statistically significant differences across computational and manually derived data by material type. However, due to increased variability in the computational data, challenges remain with identifying statistically significant differences between material types using the computational dataset. Higher resolution SEM data from particle observations with increased particle-background contrast for each material type would likely improve machine learning algorithm measurements, allowing better determination of statistically significant differences. Additional training of the machine learning algorithm would also improve the overall discrimination of particles within lower-quality observation data and improve the consistency of computationally generated measurements.

To further verify the feasibility of AI-assisted data processing to identify MNPs from samples of unknown origin, known materials were compared against the same three blinded samples using only computationally generated datasets. With cross-sectional areas generated at one-second intervals, known materials were compared against grouped suspect particles from the same three validation samples following the aforementioned methodology. Based on analyses of the 120 unknown particles studied across the three blinded validation samples, 25 and 11 unknown particles exhibited suspect deformation behavior for PET and PVC, respectively. Of the total suspect PET particles, 9, 1, and 15 were present in blinded validation samples #1, #2, and #3, respectively. Of the total suspect PVC particles, 4, 0, and 7 were present in blinded validation samples #1, #2, and #3, respectively. 

Suspect particles in each validation sample were again grouped and compared for statistical similarity to known deforming plastics PVC and PET using two-way RM-ANOVA with the computational datasets. Upon comparison, no statistically significant difference was found between the computational datasets of known PET and the grouped suspect PET particles from all three individual validation samples, accurately suggesting the possible presence of PET MNPs in all three validation samples. The same comparison between the computational datasets of known PVC and grouped suspect PVC particles from individual validation samples showed no statistically significant difference between known PVC particles and suspect PVC particles from both validation samples #1 and #3. As previously tabulated in Table 1, PVC was only present in validation sample #3, meaning that the statistical comparison of computational datasets for known and suspected PVC correctly indicated the presence of PVC in validation sample #3 but incorrectly suggested the likelihood of PVC in validation sample #1. To summarize these comparisons, it was possible to accurately predict the presence of PET particles in all three validation samples using only the computational datasets, but the presence of PVC was inaccurately suggested for validation sample #1. This analysis of computational datasets suggests that it is feasible to use computational datasets solely to predict the presence of MNPs in samples of unknown origin. However, AI detection systems would likely need to be trained with larger datasets to ensure more accurate predictions. A table of these comparisons is provided in Table 3.

## 4. Discussion

As noted earlier in this study, the random sizes and shapes of particles were expected to introduce variability across particles of similar material types. A regression analysis of the starting cross-sectional area of deforming plastic particles against the percent of total deformation by the end of observation indicated a poor correlation with an R^2^ value of 0.05. The effects of irradiation are a well-studied topic in the field of polymeric materials design [27]. Other research has suggested that the degree of deformation from electron beam irradiation also depends on the polymer’s structure [32]. Previous research has also shown that irradiation can cause a host of chemical changes, including chain scission and cross-linking in polymers, leading to overall structural changes and spatial rearrangement [23,27]. When considering the degree of crystallinity values of the tested materials, the deformation behaviors of the studied plastics matched predictions made prior to investigation, with PVC deforming the most, followed by PET and HDPE, respectively. Crystallinity is considered a significant property in the design of polymer material and was shown to have a positive correlation to the deformation and shrinkage behavior observed during irradiation in this study.

This same logic may also explain why HDPE was able to withstand the same irradiation as PVC and PET without undergoing the same deformation, which could be attributed to its already high degree of crystallinity. A plastic’s crystallinity degree is based on the density of tightly packed folded molecular chains, or crystalline lamellae regions, versus the amorphous regions where molecular chains are loosely or irregularly formed within the polymer structure [33]. Materials with higher degrees of crystallinity are associated with higher melting temperatures and greater strength and rigidity. Ranges noted in the polymer literature list degrees of crystallinity for PVC, PET, and HDPE as approximately 10%, 35%, and 75%, respectively [34,35]. Considering these concepts, increased deformation of MNPs under electron beam irradiation shows an inverse relationship to the degree of crystallinity, wherein highly crystalline MNPs would require more irradiative energy to experience significant observable deformation behavior.

The study of the plastic materials at different beam currents also allowed for the observation of differing levels of deformation, specifically with the PVC particles. By studying the same materials under both 33 nA and 37.9 nA currents, it was possible to observe how an increase in irradiation could increase particle shrinkage. Additionally, observing the non-plastic media under the higher irradiative parameters used in this study further accentuated the differences in their deformation profiles compared to plastics. In future developments of this methodology, observing the deformation of each plastic type under increasing levels of irradiation could further enhance distinctions between the deformation profiles of MNPs and non-plastic environmental media.

While the manual measurement techniques used in this investigation were enough for a proof of concept, the ultimate vision for this methodology would be to apply more computational methods for rapidly assessing the visual data collected during SEM. Using the machine learning algorithm implemented in this study on additional samples of known plastics, developing an MNP deformation database would be feasible. The development of such a tool could allow scientists and researchers to take a given environmental sample and observe particle deformations in real-time with a software package capable of rapidly estimating the presence of MNPs. Although this methodology requires access to SEM equipment, which can be costly to operate, it provides a simple, more accessible methodology to augment existing analytical techniques used in detecting environmental MNPs. SEM also has the added benefit of allowing observation of the size and occurrence of particles. 

The fields of AI, and more specifically, deep learning, are rapidly evolving. New neural network architectures and training strategies emerge yearly, achieving enhanced state-of-the-art performance in various tasks. While it is possible to improve accuracy using the latest methods or include more annotated data, accuracy can also be improved in other ways. By analyzing the spatiotemporal changes of the particles observed in this study, it is possible to leverage other well-studied tasks of AI systems, such as video classification. Classifying particle deformation into categories by polymer type and source would simplify the post-processing step of fitting the new sample to the established deformation profile of a given plastic polymer type.

The rationale for performing blinded validation studies in this investigation was to test the concept of using irradiation-induced deformation to detect MNPs in a sample with unknown media, as would likely be the case when studying an environmental sample. The results of applying the methodology described herein can potentially improve the detection of environmental MNPs. Additional studies using this methodology on less pristine and weathered samples of commercial plastics would further test its utility. Weathered or aged MNPs are expected to have varying degrees of UV radiation exposure that may impact the degree of crystallinity of aged materials. Understanding how different levels of UV exposure affect plastic particle deformation is critical in developing this methodology for use on environmental MNPs. Using similar methods to those described in this study, the deformation behavior of pristine and UV-irradiated particles of the same polymer type could be characterized and compared for differences. Under such circumstances, the analytical capabilities of the aforementioned computational tools would be necessary to make this a desirable methodology to the broader environmental research community. 

Based on this work, some key aspects of data capture that should be considered include the beam settings of the SEM instrument, the number of particles present in a sample, and the presence of media-obscuring particles. Having the correct SEM settings enables better particle resolution while aiding in observing particle morphology during deformation. Diluting samples before preparation on SEM specimen mounts can prevent particles from being obscured by large piles of microscopic debris, which aids in the more rapid collection of particle deformation data. In addition to environmental debris, the presence of biological media coating particles would likely inhibit the ability to find them using this methodology. Digestion of biological media would be recommended if this methodology is applied to an environmental sample. Guidelines for aiding in the analysis of collected observational data include having clear contrast between particles and their background, identifying isolated particles within the SEM imaging frame, and maintaining a consistent frame rate in exported video files. Following these best practices enhances the quality of observational data, enabling machine learning algorithms to better assess potential deformational changes in irradiated particles. Table 4 summarizes how sample preparation, data collection, and data analysis could be improved to make this methodology more reliable for future researchers.

Successful implementation of this method would help highlight the extent to which plastic persists in the environment, paving the way for a more comprehensive understanding of the risks associated with plastic pollution. Identifying nanoplastics by polymer type in complex environmental matrices is the ultimate validation of this methodology, providing a novel approach to help close the methodological gap for studying environmental MNPs.

## Figures and Tables

**Figure 1 micromachines-14-01903-f001:**
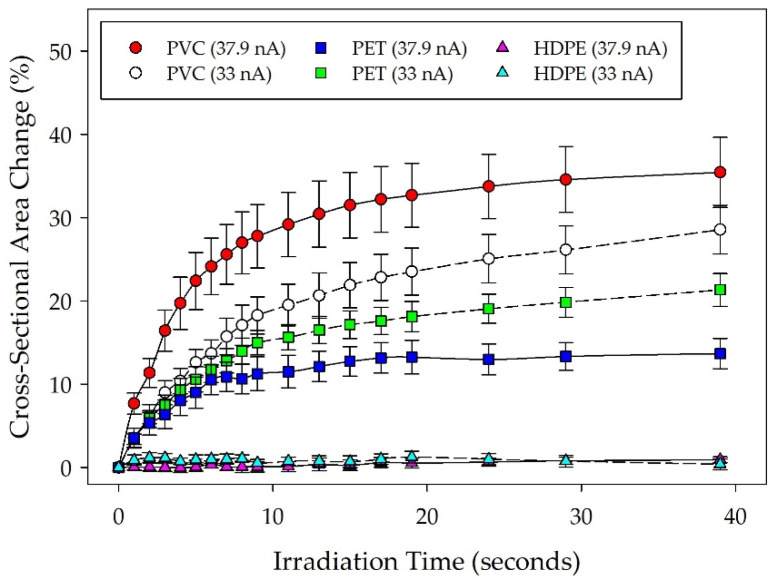
Deformation profiles for PVC, PET, and HDPE under both 33 nA and 37.9 nA beam currents. Particle shrinkage is depicted as a change in manually measured particle cross-sectional area over time, normalized against the initial particle cross-sectional area at the start of irradiation. The plot shows the mean particle measurements for each material type (*n* = 5) with standard error.

**Figure 2 micromachines-14-01903-f002:**
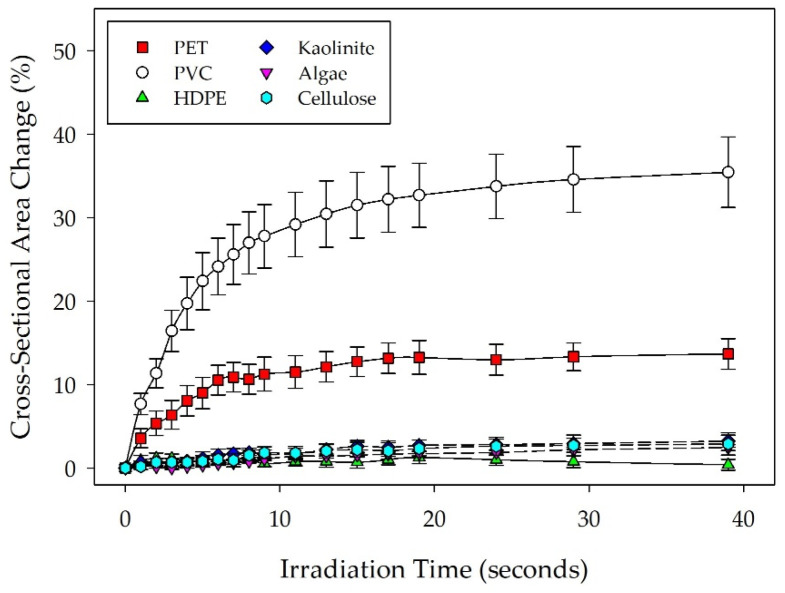
Deformation profiles for all material types under a 37.9 nA beam current. Particle shrinkage is depicted as a change in manually measured particle cross-sectional area over time, normalized against the initial particle cross-sectional area at the start of irradiation. The plot shows the mean particle measurements for each material type (*n* = 5) with standard error.

**Figure 3 micromachines-14-01903-f003:**
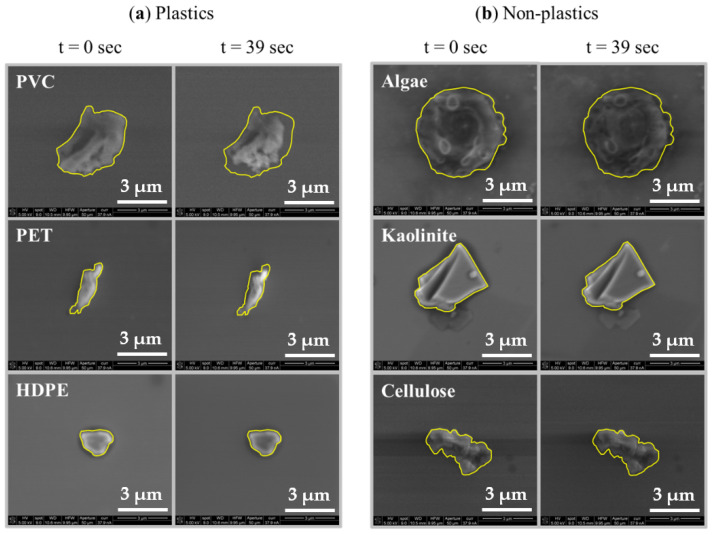
SEM images depicting typical particle deformation behavior for (**a**) plastic and (**b**) non-plastic materials from *t* = 0 s to *t* = 39 s. The highlighted boundary shows the initially measured cross-sectional area to emphasize differences over time.

**Figure 4 micromachines-14-01903-f004:**
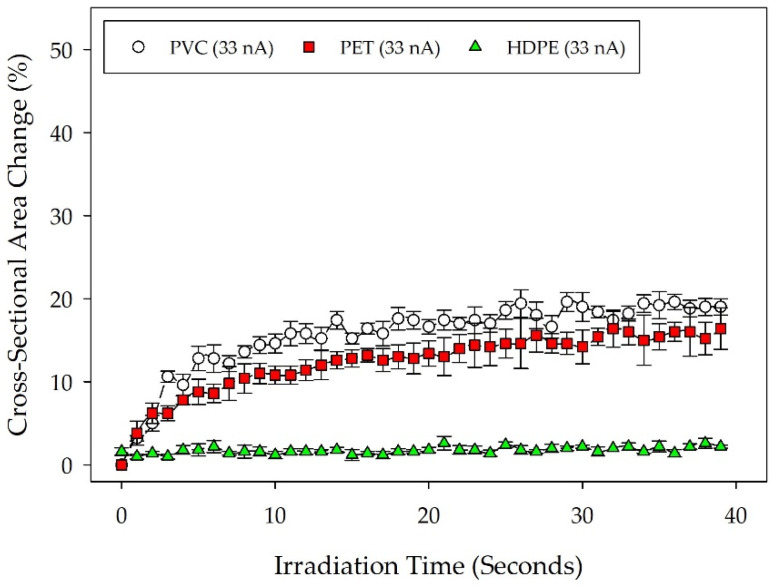
Deformation profiles for PVC, PET, and HDPE under a 33 nA beam current. Particle shrinkage is depicted as a change in computationally measured particle cross-sectional area over time, normalized against the initial particle cross-sectional area at the start of irradiation. The plot shows the mean particle measurements for each material type (*n* = 5) with standard error.

**Figure 5 micromachines-14-01903-f005:**
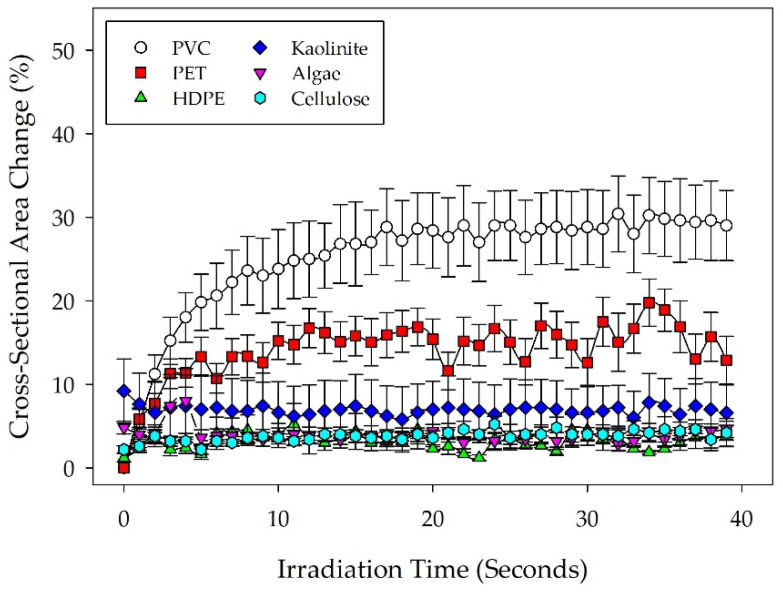
Deformation profiles for all material types under a 37.9 nA beam current. Particle shrinkage is depicted as a change in computationally measured particle cross-sectional area over time, normalized against the initial particle cross-sectional area at the start of irradiation. The plot shows the mean particle measurements for each material type (*n* = 5) with standard error.

**Table 1 micromachines-14-01903-t001:** Summarized findings of blinded validation study testing identification methodology.

Validation Sample	Statistically Suspect *	Verified	Sample Description
#1	PET	Yes	PET, Algae
#2	PET	Yes	PET, HDPE, Kaolinite
#3	PVC, PET	Yes	PVC, PET, HDPE, Silty Soil

***** denotes a significant similarity relative to known plastic media (*p* ≤ 0.05).

**Table 2 micromachines-14-01903-t002:** Statistical comparisons of manual measurements against computationally generated measurements for each material type.

Beam Current	Material Type	Significant Difference *
33 nA	PVC	No
	PET	No
	HDPE	No
37.9 nA	PVC	No
	PET	No
	HDPE	No
	Kaolinite	No
	Algae	No
	Cellulose	No

***** denotes a significant difference between measurement methods (*p* ≤ 0.05).

**Table 3 micromachines-14-01903-t003:** Summarized statistical analyses between computational datasets comparing known and grouped suspected MNPs from blinded validation samples.

Validation Sample	Statistically Suspect *	Verified	Sample Description
#1	PVC, PET	PET only	PET, Algae
#2	PET	Yes	PET, HDPE, Kaolinite
#3	PVC, PET	Yes	PVC, PET, HDPE, Silty Soil

***** denotes no statistical difference relative to known plastic media (*p* ≤ 0.05).

**Table 4 micromachines-14-01903-t004:** Guidance for data collection and analysis using SEM particle deformation methodology.

Process	Guidance	Description	Benefits
Sample Preparation	Digest	use mild digestion to remove organic matter	reduces organics surrounding MNPs to better observe particle deformation
	Fraction	separate particles of specific size ranges	improves homogeneity of particles, enabling expedited particle selection
	Dilute	reducing concentration of particles in sample	reduces aggregation, enabling expedited particle selection
Data Collection	E-Beam	optimize beam settings for voltage and current	enables observation of more discrete differences in particle deformation
	Contrast	optimize contrast settings of instrument	improves identification of particle boundaries during data analysis
	Resolution	optimize beam scan rate and image resolution	improves identification of particle boundaries during data analysis
	Selection	identification of discrete particles	improves reliability of computational methods for measuring particles
	Sample	increase number of particle observations	improves statistical power of particle deformation characterization
Data Analysis	Materials	collect data on a wide variety of materials	expands library of particle deformation behaviors, improving characterization of particles from different source materials
	Condition	collect data on particles of different condition	expands characterization of particles that have been UV aged or degraded through different processes
	Analytics	pair method with other analytical techniques	other techniques (such as EDS) may help characterize particles
	Training	provide additional training data to AI	improves computational analysis, reducing error and variability
	Software	develop SEM software package and database	enables cataloging of deformation behavior and more rapid analysis

## Data Availability

The data presented in this study are available upon request.

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
