# Peer review of "A Novel Approach for Identifying Nanoplastics by Assessing Deformation Behavior with Scanning Electron Microscopy"

_micromachines, 2023, doi:10.3390/mi14101903_

Round 1

Reviewer 1 Report

Interesting concepts matched with experimental data. 

Author Response

Thanks very much for your encouragement.

Reviewer 2 Report

Manuscript ID:   micromachines-2617247

Journal : Micromachines

Title: A Novel Approach for Identifying Nanoplastics by Assessing Deformation Behavior with Scanning Electron Microscopy

In this manuscript, the authors have represented hypothetical analysis of differentiating nanoplastics by polymer type using spatiotemporal deformation data collected through irradiation with scanning electron microscopy (SEM). The polymers are compared against proxy materials common in environmental media, i.e., algae, kaolinite clay, and nanocellulose. However, few explanations are recommended prior providing any concluding remarks related to material activity. Therefore, major revision is needed prior approval for publication.

Decision:   Major revision

1.       Being alternative to SEM, the FE-SEM works on a low voltage and allows to obtain high quality and high magnification images of nano as well as micro plastics. The authors should comment why they have not chosen FESEM for the characterization instead of SEM in this work.

Okay

Author Response

Point 1: Being alternative to SEM, the FE-SEM works on a low voltage and allows to obtain high quality and high magnification images of nano as well as micro plastics. The authors should comment on why they have not chosen FESEM for the characterization instead of SEM in this work.

Response 1: Considering that the project aim was to apply radiation energy to the particles to observe their deformation behavior, a higher voltage was preferable. Furthermore, it was also preferable to use the standard SEM to enhance the accessibility of this method for other labs to continue building upon this concept. Language to this effect has been included in the discussion pertaining to experimental design in Section 2.3 stating, “While field emission SEM (FE-SEM) may also be suitable for obtaining high resolution images of MNPs at low voltages, the higher voltages desired and the general ease of accessibility from a typical SEM was preferable”.

Reviewer 3 Report

The authors investigated a novel method to identify the nanoplastics by Scanning Electron Microscopy. The extent of global plastic pollution is extremely serious. The authors provide an interesting and novel method to identify the nanoplastics. The manuscript can be accepted after minor revision.

1.      Is this method adaptable to all kinds of nanoplastics? Otherwise, this method is only suitable for the polymers that listed in the manuscript.

2.      Why does the irradiation deform the plastic particles? Please give more explanation.

Author Response

Point 1: Is this method adaptable to all kinds of nanoplastics? Otherwise, this method is only suitable for the polymers that are listed in the manuscript.

Response 1: Per the discussion provided in Section 1 – Paragraph 5, “This study sought to test the capacity for irradiation of individual particles during SEM to provide a spatiotemporal fingerprint that would allow the identification of environmental MNPs.” All polymers are subject to irradiation damage, but the spatiotemporal response to SEM irradiation of specific polymers was tested in this study. Specific polymers with known properties were used and compared against typical media that would be present alongside environmental MNPs. Section 2.1 – Paragraph 1 provides the justification for the selection of polymer materials used in this study. Noted in Section 4 – Paragraph 1, the degree of crystallinity was consistently predictive of the spatiotemporal deformation observed in this study. The degree of crystallinity is an inherent property possessed by all MNPs. Hence, this methodology would be suitable for MNPs derived from all kinds of polymer types.

Point 2: Why does irradiation deform the plastic particles? Please give more explanation.

Response 2: Please refer to Section 1 – Paragraph 5 for the mechanisms by which irradiation may deform plastic particles. The paragraph states, “Irradiation of linear hydrocarbon polymers can result in C-C and C-H bond cleavage, radical and hydrogen removal, chain scission, cross-linking, and oxidation (in the presence of oxygen) [22]. Polymer research has shown that chemical cross-linking is an irreversible process that is commonly used to improve the strength, stiffness, and rigidity of polymeric materials [24]. These chemical alterations occurring concurrently, with high enough irradiation doses, can lead to spatial rearrangement and complex physical changes in polymer characteristics [23].” Please also refer to Section 4 – Paragraph 1 for a further explanation of irradiation deformation of polymers. The paragraph states, “The effects of irradiation is a well-studied topic in the field of polymeric materials design [27]. Other research has suggested that the degree of deformation from electron beam irradiation is also dependent on the structure of the polymer [32]. Previous research has also shown that irradiation can cause a host of chemical changes, including chain-scission and cross-linking, in polymers leading to overall structural changes and spatial rearrangement [23,27].” The cited literature in these two sections provides more detailed description of these phenomena.

Reviewer 4 Report

micromachines-2617247
Title: A Novel Approach for Identifying Nanoplastics by Assessing Deformation Behavior with Scanning Electron Microscopy
Authors: Jared S. Stine, Nicolas Aziere , Bryan J. Harper , and Stacey L. Harper
Authors aim of the present research is to adapt existing tools to address the analytical challenges posed by nanoplastics identification. In the present study, author selected polyvinyl chloride (PVC), polyethylene terephthalate (PET), and highdensity polyethylene (HDPE) to capture a range of thermodynamic properties and molecular structure encompassed by commercially available plastics. Pristine samples of each polymer type were chosen and individually milled to generate micro- and nanoscale particles for SEM analysis. To test the hypothesis that polymers could be differentiated from other constituents in complex samples, the polymers were compared against proxy materials common in environmental media, i.e., algae, kaolinite clay, and nanocellulose.
Samples for SEM analysis were prepared uncoated to enable observation of polymer deformation under set electron beam parameters. For each sample type, particles approximately 1 µm in diameter were chosen and particle deformation were recorded and studied.
Blinded samples were also prepared with mixtures of the aforementioned materials to test the viability of this method for identifying near-nanoscale plastic particles in environmental media. Based on collected experimental data, deformation patterns between plastic particles and particles present in common environmental media show significant differences. A computer vision algorithm was also developed and tested against measurements to further improve the usefulness and efficiency of this method.
Manuscript can be accepted after a minor revision with the following comments.
1.    All figures (Figs. 1,2, 4,5) are presented with black and white. It will be better to present the data with different colours.
2.    Figure 3: increase the font size of scale bar.

Author Response

Point 1: All figures (Figs. 1, 2, 4, 5) are presented with black and white. It will be better to present the data with different colors.

Response 1: The initial decision to use black and white with separate symbols was to enhance accessibility for individuals experiencing color-blindness. The symbols have been retained and color has been included to enhance the readability of the figures.

Point 2: Figure 3: increase the font size of scale bar.

Response 2: Scale bars with enlarged font size have been superimposed on the SEM images in Figure 3 to enhance readability.

Round 2

Reviewer 2 Report

The explanations are logical.

Minor gramatical editing is required